# Neurocutaneous Diseases: Diagnosis, Management, and Treatment

**DOI:** 10.3390/jcm13061648

**Published:** 2024-03-13

**Authors:** Ivelina Kioutchoukova, Devon Foster, Rajvi Thakkar, Christopher Ciesla, Jake Salvatore Cabassa, Jacob Strouse, Hayley Kurz, Brandon Lucke-Wold

**Affiliations:** 1College of Medicine, University of Florida, Gainsville, FL 32610, USA; ikioutchoukova@ufl.edu (I.K.); r.thakkar@ufl.edu (R.T.); hkurz@ortho.ufl.edu (H.K.); 2Herbert Wertheim College of Medicine, Miami, FL 33199, USA; dfost046@fiu.edu (D.F.); ccies003@fiu.edu (C.C.); jcaba091@fiu.edu (J.S.C.); jas992@cornell.edu (J.S.); 3Department of Neurosurgery, University of Florida, Gainesville, FL 32610, USA

**Keywords:** neurocutaneous disorders, cutaneous, neurologic, phakomatoses, genetic advances, precision medicine

## Abstract

Neurocutaneous disorders, also known as phakomatoses, are congenital and acquired syndromes resulting in simultaneous neurologic and cutaneous involvement. In several of these conditions, the genetic phenomenon is understood, providing a pivotal role in the development of therapeutic options. This review encompasses the discussion of the genetic and clinical involvement of neurocutaneous disorders, and examines clinical management and treatment options. With the current advances in genetics, the role of precision medicine and targeted therapy play a substantial role in addressing the management of these conditions. The interconnectedness between therapeutic options highlights the importance of precision medicine in treating each disorder’s unique molecular pathway. This review provides an extensive synthesis of ongoing and current therapeutics in the management of such clinically unique and challenging conditions.

## 1. Introduction

Neurocutaneous disorders are a group of conditions that result in long-term involvement of the nervous system and skin. Often, they are genetically inherited and symptoms emerge throughout adolescence [1]. Many of these conditions involve an understood genetic phenomenon, although spontaneous mutations can occur. These conditions also exist on a wide spectrum of phenotypes and often have multi-system involvement [1]. A challenge that occurs with diagnosing these conditions is that certain symptoms present as age progresses, further complicating the diagnosis. Given the various systems involved and the varying differences in presentation, treatment is often difficult and focuses on palliative efforts in patients with tumors or life-threatening conditions. However, given the advances in genetics, targeted therapies offer a promising future for the development of treatments in the management of neurocutaneous disorders, offering patients a tailored therapeutic option. Our review covers neurofibromatosis 1 and 2, tuberous sclerosis, Sturge–Weber syndrome, Von Hippel-Lindau disease, ataxia–telangiectasia, and Osler-Weber-Rendu syndrome, some of the diseases that are currently being studied for treatment based on their genetic phenomena.

## 2. Neurofibromatosis 1 & 2

Neurofibromatosis 1, also known as von Recklinghausen’s disease, makes up 96% of all neurofibromatosis cases and is an autosomal dominant neurocutaneous disorder caused by a germline mutation in the *NF1* gene [2,3]. NF1 affects 1 in 3000 to 4000 individuals worldwide [4]. The mutation rate of *NF1* is one of the highest in the human genome, with about 50% of cases occurring due to de novo mutations [5]. The *NF1* gene is a tumor suppressor gene on chromosome 17q11.2 that promotes the production of neurofibromin, a protein that helps regulate the cell cycle by inhibiting RAS/MAPK and PI3K-AKT-mTOR signaling pathways (Figure 1) [6,7]. The NF1 pathway has also been shown to demonstrate the two-hit pattern, in which patients inherit one mutated copy of *NF1* gene through a germline mutation and develop the phenotype of NF1 once a second hit occurs, a somatic mutation [4]. The *NF1* pathogenic variant leads to the production of non-functional or inadequate amounts of neurofibromin, which is associated with an increased risk of various tumors [8]. Neurofibromin also contains other domains, and some studies suggest that these domains play a role in the clinical manifestations of NF1. Therefore, further insight into how these domains interact and function might elucidate the clinical development of NF1 and provide a new avenue for therapeutics [9]. 

Mutations in the *NF1* gene may result in a non-functional neurofibromin protein. Neurofibromin is a tumor suppressor that downregulates the RAS/MAPK and PI3K-AKT-mTOR signaling pathways, which play a major role in cell proliferation. When these pathways go unchecked, the risk for tumor formation increases, resulting in the development of tumors in patients with neurofibromatosis 1.

Neurofibromatosis 1 presents clinically with neurofibromas; café-au-lait spots; Lisch nodules; freckles; bone abnormalities, like bowing of the legs or curvature of the spine; learning disabilities; and an increased risk of tumor formation [10,11]. Refer to Table 1 for the diagnostic criteria. NF1 can also present with peripheral neuropathy. Patients can vary in their presentation, but one study has shown that patients can develop sensory and/or motor manifestations. The study determined that peripheral neuropathies are a serious consequence of NF1 and are associated with mortality due to spinal complications [12]. Spinal neurofibromatosis is another form of NF1 that is defined as bilateral neurofibromas, which involve all spinal roots, resulting in spinal nerve compression. These symptoms manifest as pain, weakness, or sensory changes; therefore, early detection and management are critical [13].

Additionally, neurofibromatosis 1 can present with mosaicism and cutaneous manifestations [14,15]. Notable exceptions include NF-1 patients with microdeletions, who make up 5% to 11% of all NF-1 patients. This subgroup of patients often presents with tall stature and dysmorphic facies, along with severe global development delays and cognitive disability presenting at an earlier age. Patients with microdeletions are at an increased risk of malignant peripheral nerve sheath tumors. Therefore, early genetic testing is crucial for managing patient treatment plans [16]. Genetic testing has been gaining significance as a tool to distinguish neurofibromatosis 1 from other conditions and identify mosaicism in patients [8].

**Table 1 jcm-13-01648-t001:** Diagnostic criteria of NF1.

Diagnostic Criteria of NF1
Six or more café-au-lait macules over 5 mm in diameter in prepubertal individuals or over 15 mm in postpubertal individuals Freckling in axillary of inguinal regions Two or more neurofibromas of any type or one plexiform neurofibroma Optic pathway glioma Two or more iris Lisch nodules, or two or more choroidal abnormalities Sphenoid dysplasia, anterolateral bowing of the tibia, or pseudoarthrosis of long bone Heterozygous pathogenic *NF1* variant with a variant allele fraction of 50%

Diagnosis of NF1 requires at least two of the following features in an individual without an affected parent, or at least one of the following features in a parent with NF1 [17].

Current treatment aims to prevent complications and improve patient quality of life. Regular monitoring is recommended for tracking the progression of the patient’s condition and to detect any complications early. Monitoring includes blood pressure checks, spinal checks, and whole-body MRIs to detect malignant transformations. Surgical management of painful or uncomfortable neurofibromas may not prevent the recurrence of neurofibromas, but may provide the patient with pain relief and improve quality of life. Selumetinib is the first FDA-approved treatment for inoperable plexiform neurofibroma in patients with neurofibromatosis 1 that seeks to induce anti-tumor activity by inhibiting the MEK pathway [18,19,20]. Patients taking selumetinib have reported improved physical function, decreased pain, better mobility, and overall mental health. Other MEK inhibitors (binimetinib, mirdametinib, and trametinib) are also currently being investigated for NF1 plexiform neurofibromas [21]. Additionally, research is being done to target the other domains of neurofibromin, along with possibly looking into vectors to deliver DNA [22].

Neurofibromatosis 2-related schwannomatosis is an autosomal dominant neurocutaneous disorder caused by mutations in the *NF2* gene on chromosome 22q12, which has been shown to follow a two-hit pattern. The *NF2* gene codes for the formation of the protein, Merlin (schwannomin), a tumor suppressor that is associated with the formation of central nervous system tumors upon *NF2* gene mutation [23,24]. NF2 affects 1 in 25,000 to 40,000 individuals [25]. Approximately half of neurofibromatosis 2 cases occur because of spontaneous mutations, despite being an autosomal dominant disorder [26]. Like neurofibromatosis 1, neurofibromatosis 2 can exhibit mosaicism and symptoms can localize in regions of affected cells. The severity of the disorder is variable and can present early on in life with multiple tumors, or later in life with a milder presentation [27].

Cutaneous manifestations of neurofibromatosis 2 are less prominent compared to neurofibromatosis 1, and do not commonly include the dermatological findings normally seen in neurofibromatosis 1. Distinct features of neurofibromatosis 2 mainly include the development of tumors that arise from Schwann cells, like vestibular NF2-related schwannomas that can be associated with tinnitus, hearing loss, and balance dysfunction. Meningiomas and ependymomas are also common [28]. Additionally, neurofibromatosis 2 may present with cataracts, retinal abnormalities, and, rarely, cutaneous NF2-related schwannomas under the skin [2].

The less prominent cutaneous manifestations of neurofibromatosis 2 help distinguish it from neurofibromatosis 1 clinically. The diagnosis of neurofibromatosis 2 mainly involves clinical presentation of the associated tumors, family history of NF2, and genetic testing [29,30].

While neurofibromatosis 2-associated tumors are largely benign, lack of treatment can lead to auditory, facial, and vestibular functional deficiencies, and extreme cases can lead to brainstem compression, obstructive hydrocephalus, and death. Surgery and radiation are treatment options that pose major risks, like hearing loss, intracranial bleeding, and stroke. Additionally, surgical removal is not possible in multiple situations where tumors commonly arise in patients with neurofibromatosis 2, and radiation treatment may transform or accelerate the growth of NF2-related schwannomas. As a result, neurofibromatosis 2 does not have a curative or long-term therapy [31,32].

Emerging therapeutic developments are drugs like bevacizumab, endostatin, and axitinib, which seek to downregulate VEGF expression. Bevacizumab has shown promising results in tumors that grow rapidly and was also noted to improve hearing in 20% of patients [33]. Antisense oligonucleotides targeting exon 11 demonstrated promising results in in vitro studies, but delivery of the drug and effective formulation remain challenges. Other approaches include RTK-targeting drugs, PI3K/Akt/mTORC1, Ras/Raf/MEK/ERK signaling pathway-targeting drugs, and the use of aspirin. Additionally, therapies directed against tumor-associated macrophages are a potential avenue of investigation [34]. Trials are ongoing and more research is needed to determine the efficacy of treatment [34,35].

## 3. Tuberous Sclerosis

Tuberous sclerosis complex (TSC) represents a neurocutaneous genetic disorder characterized by the development of benign tumors, known as hamartomas [36,37]. TSC is estimated to impact between 1 in 6000 to 10,000 individuals [38]. The clinical expression of TSC arises from disruptions in various cellular functions. TSC results from autosomal dominant or sporadic mutations in the *TSC1* or *TSC2* genes, encoding the hamartin and tuberin proteins, respectively [37,39,40,41,42]. These proteins jointly regulate cell proliferation through the mammalian target of the rapamycin pathway (mTOR). Mutations in either gene lead to dysregulation of the TSC1:TSC2 complex, consequently affecting the mTOR pathway and causing tissue overgrowth [43]. Although inherited in an autosomal dominant manner, it is important to note that around 70% of TSC cases result from de novo mutations in the germline [44].

Clinically, TSC impacts multiple organ systems (Figure 2), with a predilection for the central nervous system (CNS), skin, and kidneys [36]. Most common CNS symptoms encompass seizures, subependymal nodules (SENs), and subependymal giant cell astrocytomas (SEGA) [45]. Renal complications arise from renal angiomyolipomas, which are benign tumors forming in the kidneys. Lastly, cutaneous manifestations involve hypomelanotic macules, adenoma sebaceum, shagreen patches, and ungual fibromas [36].

Multisystemic manifestations of TSC include the nervous system, skin, lungs, heart, kidneys, and eyes, presenting variable phenotypes.

Regarding dermatologic abnormalities, approximately 90% of patients develop hypomelanotic macules, also known as ash leaf spots [36,46,47]. These are oval-shaped characteristics that are usually present at birth, and a Wood’s lamp examination improves its detection [48]. Adenoma sebaceum, formerly known as facial angiofibroma, manifests as small hyperpigmented macules in a butterfly pattern on bilateral cheeks [48]. Shagreen patches are leather plaques with an orange-peel appearance due to slightly depressed hair follicles [48]. Ungual fibromas are commonly observed periungually and subungually, particularly in toenails, emerging later in puberty [48].

Diagnosing TSC relies on a constellation of features rather than a singular symptom. Definitive diagnostic criteria necessitate two major features, or one major and two or more minor features (Table 2). Major features observed more frequently in TSC patients include 11 clinical findings uncommon in the general population [49]. Minor features, also more prevalent in TSC patients, are common in the general population [49]. Additionally, imaging studies, such as CT and MRI, are crucial for assessing CNS manifestations. Genetic testing is also conducted to identify mutations in the TSC1 or TSC2 genes.

Treatment options for TSC-associated skin lesions encompass both non-pharmacological and pharmacological approaches. Systemic mTOR inhibitor treatment is indicated in patients with multisystemic manifestations, addressing the underlying TSC pathophysiology and potentially improving dermatologic symptoms [49]. Alternatively, topical mTOR inhibitors or surgical procedures may be considered in patients not receiving systemic treatment [49].

Notably, in 2022, the Food and Drug Administration (FDA) approved HYFTOR (sirolimus) gel, the first FDA-approved topical intervention designed for individuals with TSC-presenting facial angiofibroma [50]. Lastly, diverse surgical modalities are available, including laser-based procedures. For elevated angiofibromas, ablative lasers, such as carbon dioxide (CO2) or erbium: YAG laser, are frequently used [51]. Additionally, a vascular laser, capable of selectively obliterating blood vessels with minimal scarring risk, can be employed for the treatment of flat red spots [51].

## 4. Sturge–Weber Syndrome

Sturge–Weber syndrome (SWS) is a rare neurocutaneous disorder, distinctively marked by port-wine vascular macules and café-au-lait spots on the skin and eyes [52,53,54]. Unlike many genetic conditions, SWS is not hereditary but stems from a sporadic mutation in the *GNAQ* gene, leading to a range of abnormalities in the eyes, skin, and brain with varied clinical presentations, from asymptomatic to severe [55,56].

The underlying mechanism involves a postzygotic somatic mutation in *GNAQ* on chromosome 9q21.2, sometimes also implicating the homogenous *GNA11* gene on chromosome 19p13.3 [55]. These mutations create a defect in the Gαq protein that does not allow it to properly shut off, contributing to conditions like phakomatosis pigmentovascularis through mosaic expression [57,58,59]. SWS’s port-wine lesions result from excessive capillaries around the trigeminal nerve in the face and abnormal brain vessels that can lead to tissue atrophy and tram-line calcification [60,61,62,63]. Diagnosis primarily relies on clinical observation and imaging studies, like MRI and CT scans, which reveal the extent of cerebral involvement [62].

Neurologically, seizures are a significant concern in SWS, often correlating with the size of leptomeningeal angiomas [64]. The onset age of seizures is crucial for prognosis, with early-onset seizures linked to more significant neurological challenges [65]. Glaucoma, another critical aspect of SWS, tends to emerge in early childhood and can lead to vision loss if not managed promptly. The variable nature of glaucoma progression and visual outcomes underscores the importance of early intervention [66,67].

The prevalence of autism spectrum disorders and communication difficulties in SWS patients highlights the need for comprehensive neurodevelopmental assessment [68]. Furthermore, the disorder can lead to a range of developmental and neuropsychiatric comorbidities, significantly affecting psychological functioning, especially in younger patients [69].

SWS management requires a personalized, interdisciplinary approach. Laser therapy, particularly pulsed dye laser, is widely recognized for treating port-wine stains [70]. In terms of neurological complications, anti-convulsant therapy is crucial for seizure management, tailored to the seizure type and patient’s age [71]. For refractory cases, surgical options like hemispherectomy might be necessary, although they carry significant risks [67,72]. Glaucoma management often involves a combination of medical and surgical interventions, with regular ophthalmological assessments being essential for early detection and treatment [68]. Additionally, addressing neurodevelopmental and psychological aspects through early intervention programs and neuropsychological evaluations is vital for managing developmental delays, learning disabilities, and neuropsychiatric issues [52,69].

Recent discoveries, particularly the identification of the *GNAQ* gene mutation, have improved our understanding of SWS and opened possibilities for targeted molecular therapies (Figure 3) [52,73]. Advances in MRI technology have enhanced the visualization of leptomeningeal angiomas, aiding in early diagnosis and informing treatment decisions, especially concerning epilepsy management [74]. However, treatment strategies, especially regarding epilepsy and cutaneous symptoms, remain subjects of debate. Discussions are ongoing about the timing, aggressiveness of seizure management, and balance between medication side-effects and seizure control [75]. Surgical interventions in intractable cases and the optimal timing and type of laser therapy for port-wine stains are also debated, considering the risks of skin damage and the psychological impact [76,77].

Sturge-Weber syndrome results from a sporadic mutation in the GNAQ gene. This mutation leads to impaired function of the Gαq protein, leading to excessive capillary proliferation in a CN V1/V2 distribution on the face. This results in the classic port-wine stain presentation.

## 5. Von Hippel-Lindau Disease

Von Hippel-Lindau (VHL) disease is a rare genetic disorder that causes a variety of tumors and cysts to manifest throughout various organ systems [78]. These tumors can manifest as benign or malignant, and show commonly in the central nervous system, kidneys, adrenal glands, and pancreas. Most notably, Von Hippel-Lindau disease was found to be the first renal cancer disorder with a defined genetic basis [79]. Specifically, this relates to inherited renal cell carcinoma. Another common manifestation of the disease is hemangioblastomas, especially those in the eye [80]. There are two different types of VHL dependent on the presence of pheochromocytoma and the additional presence of renal cancers. Pheochromocytomas are benign tumors typically found in the chromaffin cells of the adrenal medulla or paraganglion [81]. Type 1 VHL is classified via the absence of pheochromocytoma. VHL type 2 is characterized by the presence of pheochromocytomas and is further broken down into three subtypes. Type 2A is with renal cancer. Type 2B is without renal cancer. Type 2C is individuals who develop pheochromocytomas only (Figure 4) [82].

Von Hippel-Lindau is an autosomal dominant disease. VHL is rare and found in about 1 in 36,000 people, and it has a de novo manifestation rate of about 20% in new cases [83,84]. It is an inherited germline mutation of the *VHL* gene on chromosome 3p25 [85]. This leads to a mutated end product of the *VHL* gene, the VHL protein (pVHL). This protein causes the degradation of hypoxia-inducible factor (HIF), which is responsible for oxygen regulation in cells [78]. When this protein is degraded, HIF is uncontrolled and upregulated, along with other growth factors promoting tumor formation. Platelet-derived growth factor (PDGF) and vascular endothelial growth factors (VEGFs) contribute significantly to tumor formation [83]. HIFs encourage an angiogenic state of continuous mitogenic signaling [86]. Overall, this combination of growth factors leads to an environment where tumors can grow unchecked and cause a variety of lesions throughout the body.

It is important to note that Von Hippel-Lindau disease has a variety of cutaneous manifestations. While most of the effects of the disease are internal, about five percent of cases do show significant cutaneous features [87]. However, VHL is a phakomatosis with very few and sporadic cutaneous findings. These are seen as melanocytic nevi, café-au-lait spots, and capillary malformations [88]. Melanocytic nevi (non-cancerous moles) are a common benign neoplasm of the skin. They appear on the skin as darkened non-cancerous moles, and affect the melanocytes responsible for skin pigment synthesis [89]. This is important to consider in patients showing this cutaneous manifestation of the disease as melanomas arise from nevi about 25–33% of the time [90]. Café-au-lait spots are hyperpigmented spots that are dark to light brown and often appear at birth or early in life [91]. Typically, they are flat and vary in size. The presence of multiple spots is often linked to genetic syndromes, as seen in the association with VHL. While these are uncommon findings in patients with Von Hippel-Lindau disease, they are a key feature that puts them in the category of neurocutaneous diseases and can help in the diagnosis and treatment of patients. It is unclear whether the type of VHL determines the likelihood of developing cutaneous manifestations.

The diagnosis of Von Hippel-Lindau disease uses a mix of clinical criteria involving manifestations and genetics/family history of the disease, along with genetic testing to confirm a diagnosis. There are three main clinical diagnostic criteria used in proband: Dutch, Danish, and International [92]. Each varies slightly in what they utilize for a family history of VHL or a related tumor and individual diagnosis of a VHL-related manifestation [93]. The Dutch and International criteria require one VHL-related tumor and a first- or second-degree relative diagnosed with VHL or with more than one VHL-related tumor. The Danish criteria is stricter in that it only requires a relative to the first degree to be positive for VHL [93]. If no known family history is available, then the Dutch criteria requires two or more VHL-related manifestations to be present. The Danish and International criteria require either two or more hemangioblastomas or one hemangioblastoma and one other VHL-related manifestation to be present to be considered for this disease [93]. Molecular diagnosis is used to find a heterozygous pathogenic variant of *VHL* and is a conclusive way to obtain a diagnosis [78]. The two types of genetic testing are gene-targeted testing and comprehensive genomic testing. This includes single and multigene panel tests. Single gene testing involves a sequence analysis of the VHL coding region and flanking sequences. It is used to detect variants, insertions, or deletions. A multigene panel is performed and includes VHL, along with other genes of interest. This panel is used if the patient presents with symptoms resembling another disease. These criteria are the standard combination for assessing an individual’s likelihood of having the disease and follow through with genetic testing to confirm a diagnosis [92].

The management and treatment of Von Hippel-Lindau disease varies in each case as the manifestations of the disease vary. There is no standard or universal treatment for VHL, but options exist for the varying symptoms that may arise. Targeted therapies for advanced renal cell carcinoma include the oral treatment Pazopanib, which inhibits VEGF, PEGF, and the stem-cell factor receptor c-kit [94]. Similarly, belzutifan is a newly FDA-approved, oral drug that is used for the treatment of renal cell carcinoma in VHL. It is an inhibitor of HIF-2α and has been very effective in treatment [95]. Renal cell carcinoma can also be treated using radiofrequency ablation or cryoablation over surgical resections [96]. Most tumors in VHL are safe to be removed. Central nervous system hemangioblastomas and selected spinal hemangioblastomas are safe and allow for full tumor resection [97,98]. Retinal hemangioblastomas are treated early as they can lead to vision loss. Treatment is safe and does not cause damage to eyesight. Treatment of retinal angiomas includes diathermy, xenon, laser, cryocoagulation, and external beam radiotherapy [78]. Pheochromocytomas should be removed laparoscopically and it is recommended to watch pheochromocytomas that are less than two centimeters in size as they are not considered harmful [92]. Surveillance is recommended for the common manifestations of VHL as preventative care and the early finding of tumors can help to improve patient outcomes. Lastly, it is important to consider genetic counseling in VHL patients who wish to have children. The disease is autosomal dominant, meaning an affected individual has a 50% chance of passing the mutation to offspring [92].

VHL can be broken into two main types based on the presence of pheochromocytomas. Type 2 has three subtypes based on the presence of renal cancers.

## 6. Ataxia–Telangiectasia

Mutations in the *ATM* gene result in a defective ataxia–telangiectasia mutated serine/threonine kinase. Functioning ATM protein acts as a tumor suppressor, with vital roles in the progression of the cell cycle, induction of apoptosis, and repair of double-stranded DNA breaks through paramount interactions with p53, CHK2, and ABL. Downstream effects of the deficient *ATM* result in the inability to correct damage in the genome, hindering the production of mature T and B cells, leading to immunodeficiency and increased susceptibility to the development of certain malignancies.

In the discourse of inherited neurological disorders, ataxia–telangiectasia (AT), also recognized as Louis–Bar syndrome, remains a staple of discussion with its characteristic cutaneous manifestations and etiology [99]. Fortunately, as a neurodegenerative disorder, AT is seldom diagnosed, occurring in 1 out of 100,000 births in the United States, with some populations having an increased incidence of up to 1 out of every 40,000 births [100]. Ataxia–telangiectasia is inherited in an autosomal recessive manner and remains one of the most debilitating primary immunodeficiencies, with patients generally dying before the third decade of life [101]. However, AT’s prognosis and comprehension have undergone significant expansion with the advent of genetic testing, paired with an increased understanding of its clinical presentation and pathophysiology.

This autosomal recessive disorder’s morbidity is directly linked to its deleterious involvement in multiple organ systems within the body [102]. Arising from mutations in the ATM gene, its viral roles in the repair of damaged DNA can lead to a variety of manifestations, such as the gradual degeneration of the cerebellum leading to progressive ataxia, the presence of cutaneous telangiectasias, an elevated risk of malignancies (especially lymphoid malignancies), compromised immune function, and frequent sinopulmonary infections (Figure 5) [103]. Specifically, a mutation located at 11q22–23, which encodes for the *ATM* gene, results in a defective Ataxia–Telangiectasia Mutated Serine Threonine Kinase (ATM) protein that insufficiently addresses double-stranded breaks in DNA [104]. This impaired response is vital in the presence of external stressors, such as ionizing radiation or chemicals, as well as the development of mature T cells [105]. The subsequent faulty ATM protein is unable to maintain critical interactions with p53 and ABL, leading to the inability to induce apoptosis and perform DNA repair by homologous recombination repair [106]. Additionally, CHK2′s interactions with CDC25A and CDC25C contribute to the bypassing of cell cycle checkpoints at S and G2-M, further exacerbating the loss of p53 bypassing the G1-S checkpoint [107].

Ataxia–telangiectasia is an autosomal recessive disorder due to mutations in the *ATM* gene on chromosome 11q22. The *ATM* gene encodes an ATM kinase involved in detecting DNA damage and regulating the cell cycle. Mutations fail to induce apoptosis during the G1-S transition and cause an inability to repair DNA, resulting in cell cycle progression.

One of the distinctive features prompting clinicians to diagnose AT is its characteristic cutaneous features, most notably the telangiectasias [108]. Telangiectasias are identified by the presence of smaller, vasodilated blood vessels on the skin, frequently associated with dilation or broken blood vessels located near the surface of the skin or mucous membranes [109]. In the progression of AT, telangiectasias are observed on the whites of the eyes and various areas on the skin, particularly the bulbar conjunctiva, ears, neck, and cubital fossa [110]. The exact cause of telangiectasia remains unknown; however, researchers propose that the development of these manifestations includes weakening of the blood vessel structure and function, stemming from genetic, environmental, or an interplay of these influences [103]. It is widely accepted that there is a consensus in acknowledging AT patients exhibit variability in the cutaneous manifestation of the disease [111]. The degree of telangiectasias can vary influenced by aforementioned factors; patients may present with extensive telangiectasias, while others may not manifest them at all [112].

Diagnosing AT presents a multifaceted challenge from the absence of a consensus on clinical diagnostic criteria, requiring a comprehensive evaluation of clinical features to confirm [113]. Arising from the diverse array of affected organs, as well as the varying severity observed in patients, the complexity of diagnosing patients accounts for the combination of neurologic clinical features in tandem with one or more cutaneous or laboratory findings [108]. A lack of coordination and instability of movements affecting balance and fine motor skills (ataxia), the development of clusters of dilated blood vessels (telangiectasias) on mucous membranes, and frequent infections are indicative symptoms that lead toward AT diagnosis [114]. The diverse range of symptomatology underscores the importance of a comprehensive personal and family history, which can lead a practitioner toward genetic testing to identify the presence of the 11q22–23 mutation in an individual suffering from cerebellar effects [115]. Moreover, analysis of the patient’s blood may provide key insights in diagnosing AT, revealing the presence of either decreased immunoglobulins or increased alpha-fetoprotein (AFP) levels in adults [116]. A significant reduction in immunoglobulins in AT results from the lack of mature T cells and the impaired interaction activating B cells [117]. While elevated AFP levels in infants are commonplace, their decline is associated with age; however, markedly higher levels in adults are characteristic of AT, making the recovery of AFP a hallmark diagnostic tool [118].

Unfortunately, as of now, there is no cure for AT, the extent of treatment is supportive care aiming to address neurological dysfunction and slow deterioration [119]. The challenge in treating AT stems from the inherited gene mutation on chromosome 11; the lack of gene editing capabilities to correct such a mutation leaves the population in a persistent state of risk while being minimal [120]. In terms of supportive care, immunoglobulin supplementation and antibiotics are routinely utilized to proactively prevent and halt opportunistic infection from the immunodeficiency seen in patients with AT [121]. Moreover, the utilization of chest physiotherapy and airway clearance techniques in children with recurrent sinopulmonary disease have demonstrated the reduced development of chronic lung disease in AT [122]. Lastly, the administration of glucocorticoids in a prospective cohort study has demonstrated promise in relieving ataxia symptoms in AT; however, the long-term effectiveness is yet to be established [123,124]. Recent advancements in genetics and immunology have yielded transformative work, igniting a diverse range of novel treatments that holds considerable potential as innovative therapies in the treatment of AT. In an effort to correct the hematologic abnormalities seen in AT, bone marrow transplantation is becoming an increasing option of therapy to promote immune competence while preventing leukemia [125]. Recent publications have shed light on the use of splice-switching antisense oligonucleotides (ASOs) which are able to modify cellular pre-mRNA to alter and correct splice-altering genetic variants that cause AT, leading practitioners in the future to potentially use ASOs enabling correction of splice-altering genetic variants that cause disease [126]. Vector therapy has also been investigated regarding AT. Lentiviral and gammaretroviral vectors have been studied and successfully delivered the ATM gene to target cells to restore ATM function. Additionally, researchers have investigated the role of HSV-1 and HSV/AAV hybrid amplicon vectors and have shown promise in animal models. However, due to the large size of the ATM coding sequence, they might not be suitable in patients [120].

## 7. Osler-Weber-Rendu Syndrome

Osler Weber Rendu disease, also known as Hereditary Hemorrhagic Telangiectasias (HTT), is a disease inherited in an autosomal dominant manner with a later onset of symptoms [127,128]. The prevalence of HHT in the general population is reported to range from 1.5 to 2 out of 10,000. In terms of the types of HHT, HHT is divided into two main forms, HHT 1 and HHT 2 (Figure 6). The HHT 1 form is reported to be more prevalent than the HHT 2 form [127]. The two forms are initiated by heterozygous mutations [127]. HHT 1 is initiated by a mutation in endoglin, while HHT 2 is initiated by a mutation in Activin A receptor-like type 1 [127]. Both of these mutations lead to an interruption of the typical TGF beta-mediated pathways [127]. Consequently, arteriovenous malformations develop and affect the way that blood vessels form, initiating atypical connections between veins and arteries and, subsequently, avoiding passing through capillaries [127]. Arteriovenous malformations (AVMs) of the pulmonary system, brain, spine, or liver can occur in patients with HHT [127]. Visceral AVMs, such as brain AVMs, can lead to complications, such as hemorrhagic strokes and seizures [128]. Brain AVMs may also lead to motor weakness and aphasia [129]. A less common form of HHT is one with mutations within SMAD4, in which a patient subsequently exhibits traits of both HHT and juvenile polyposis syndrome [130]. 

A common manifestation of Osler-Weber-Rendu disease is the formation of telangiectasias [130]. About three-fourths of the HHT patient population display telangiectatic lesions [130]. Telangiectasias are dilated blood vessels formed in various parts of the body, both visceral and cutaneous [128]. Cutaneous telangiectasias are often described as “spider-web-like” in appearance, affecting several external surfaces of the body, including commonly the lips, tongue, and face [127]. In an article by Hyldahl et al. (2022), a more thorough description and evaluation of telangiectasias of HHT patients were performed [130]. Per the Hyldahl et al. article, the lesions of HHT patients were most commonly described as round, flat, or minimally raised/elevated [130]. More specifically, telangiectatic lesions can be categorized as confluent round, radiating round, lacunar round, tortuous round, hemangioma-like round, dilated, and arborized [130]. Overall, per the Hyldahl et al. article, HHT1-type patients are more likely to develop telangiectatic lesions, both mucosal and cutaneous, compared to HHT2-type patients. Additionally, as patients age, HHT patients are more likely to form telangiectatic lesions and acquire dilated blood vessels [130]. Additionally, over 90% of HHT patients from the Hyldahl article experienced epistaxis. Less commonly, patients may also experience bleeding from the gums, lips, ears, cheeks, and digits [130]. 

Early screening is often recommended if HHT is suspected in patients due to the severe health threat that visceral AVMs may pose [128]. Some criteria to consider when screening for HHT include family history/first-degree relative with HHT, recurrent epistaxis, visceral lesions, and the formation of telangiectasias [131]. Genetic testing for mutations in Activin A receptor-like type 1 (ALK1) and endoglin (ENG) is recommended for the diagnosis of HHT [127]. In terms of the epistaxis and subsequent bleeding disorders, such as anemia, that HHT patients often develop, patients may be treated with anti-angiogenics [128,132]. More commonly in the older HHT patient population, GI bleeding may develop and, subsequently, needs to be addressed with anti-angiogenic modalities and iron transfusions [128]. As a result, clinical recommendations for HHT patients also include iron levels and anemia testing [133]. Treatments that patients are provided regarding nasal cavity telangiectasias include bevacizumab, laser therapies, cauterizations, nasal implants, and surgical procedures to close nasal cavities [130]. In terms of the treatment of visceral AVMs, depending on the location of the lesion, patients may undergo embolization, resection, or focal radiation [128]. Specifically for hepatic AVMs, liver transplants or bevacizumab are the mainstays of treatment, since hepatic lesion embolization leads to subsequent liver issues in the future [128]. However, the full consequences of the long-term use of bevacizumab have not been fully determined at this time, as increased thrombotic risk occurs with use [133]. More research is currently inquiring about the efficacy of targeting VEGF in medications for HHT patients to inhibit angiogenesis, given that HHT patients often have an active pro-angiogenesis process [133].

This figure demonstrates the types of hereditary hemorrhagic telangiectasias based on gene mutation types and their prevalence in the HHT patient population.

## 8. Conclusions

Although the clinical characteristics and genetic basis of neurocutaneous disorders differ, the role of precision medicine and targeted therapy provides a shared theme. Emerging therapeutics for NF-1 include selumetinib, an MEK inhibitor. Antiangiogenic agents are being used for the management of NF-2, along with Osler-Weber-Rendu syndrome, to manage symptoms. mTOR inhibitors are gaining prevalence for the management of TSC. Regarding VLH, treatments focus on managing renal cell carcinoma. As these conditions are inherited genetically, genetic counseling should be performed for individuals presenting with symptoms concerning neurocutaneous disorders. Overall, the role of precision medicine and targeted therapies is increasing, elucidating the genetic phenomenon behind neurocutaneous disorders. With improved understanding, modern medicine can develop interventions to mitigate symptoms and improve patient outcomes.

## Figures and Tables

**Figure 1 jcm-13-01648-f001:**
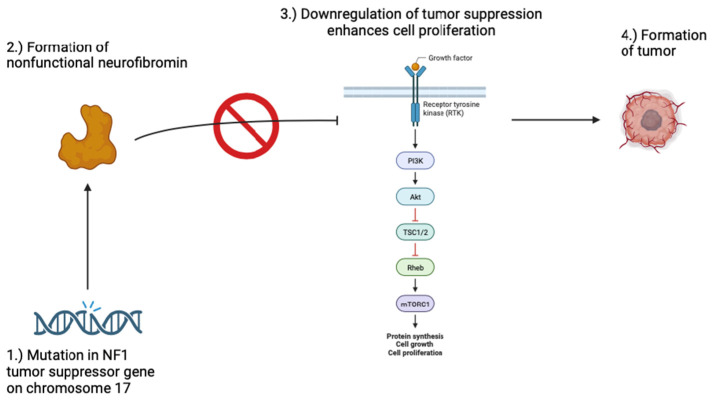
Pathogenesis of neurofibromatosis 1.

**Figure 2 jcm-13-01648-f002:**
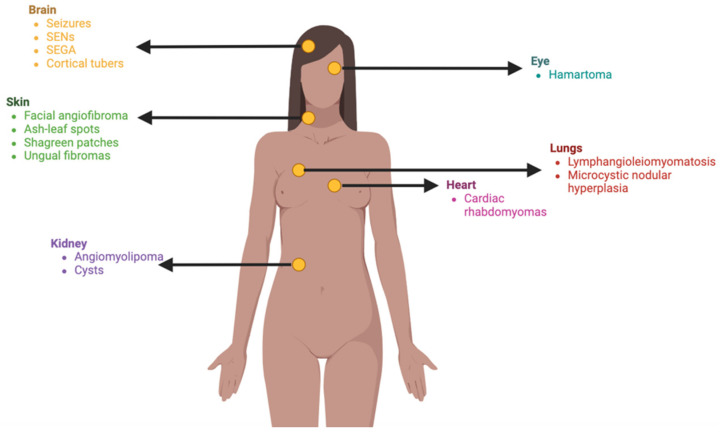
Multisystem manifestations of TSC.

**Figure 3 jcm-13-01648-f003:**
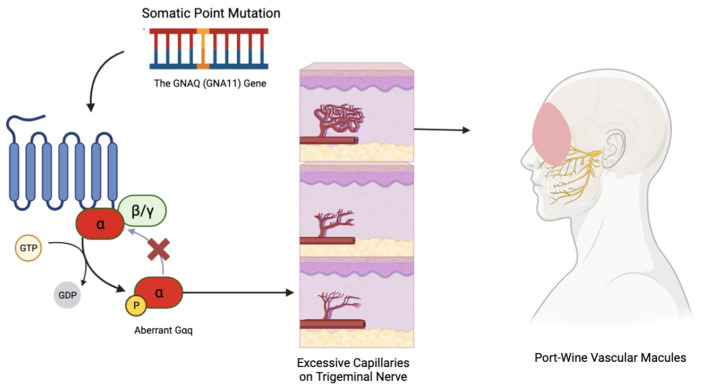
How the vascular macules manifest in Sturge–Weber syndrome.

**Figure 4 jcm-13-01648-f004:**
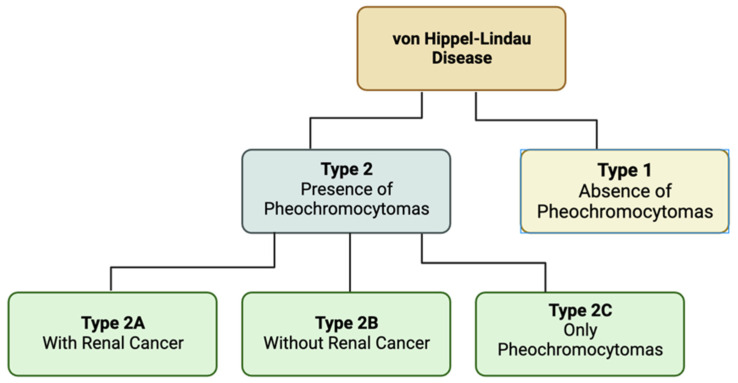
Types of Von Hippel-Lindau disease.

**Figure 5 jcm-13-01648-f005:**
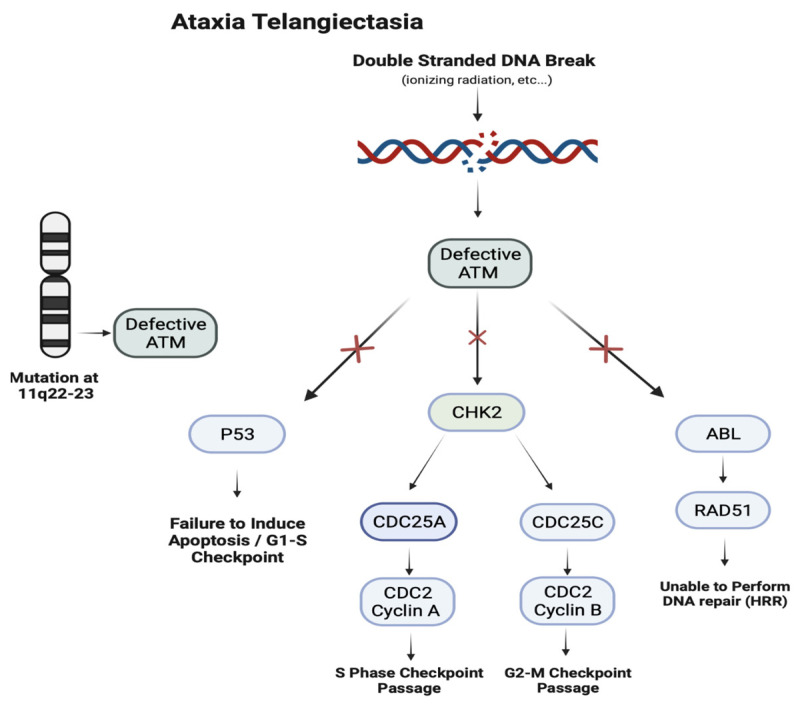
Pathogenesis of ataxia–telangiectasia.

**Figure 6 jcm-13-01648-f006:**
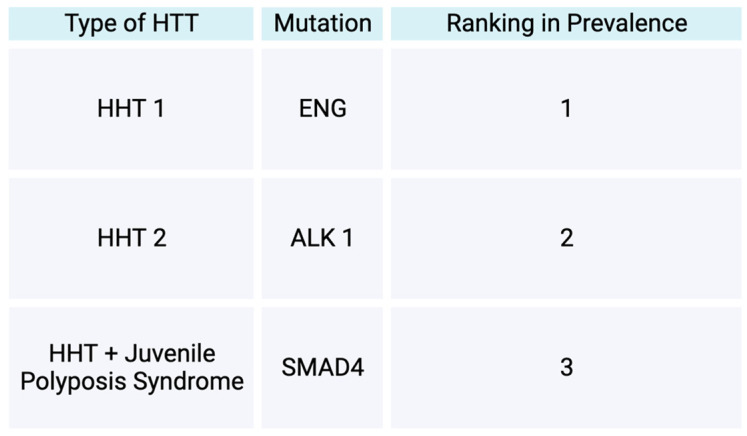
Types of hereditary hemorrhagic telangiectasias.

**Table 2 jcm-13-01648-t002:** Major and minor criteria for the diagnosis of TSC.

Tuberous Sclerosis Diagnostic Criteria	
Major	Minor
Hypomelanotic macules Angiofibromas (≥3) Ungual fibromas (≥2) Shagreen patch Cortical dysplasias Multiple retinal hamartomas Subependymal nodules Subependymal giant cell astrocytoma Cardiac rhabdomyoma Lymphangioleiomyomatosis Angiomyolipomas (≥2)	‘Confetti’ skin lesions Dental enamel pits (>3) Intraoral fibromas (≥2) Retinal achromic patch Multiple renal cysts Non-renal hamartomas

Diagnosis of TSC requires at least two major features, or one major and two or more minor features.

## Data Availability

The data presented in this review is available through the National Library of Medicine.

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
