# Peer review of "Neurocutaneous Diseases: Diagnosis, Management, and Treatment"

_jcm, 2024, doi:10.3390/jcm13061648_

Round 1
Reviewer 1 Report
Comments and Suggestions for Authors
Reviewer’s opinion
The manuscript deals with diagnosis, management and treatment of neurocutaneous syndromes.
The purpose of this review is to provide information on ongoing and current therapeutic approaches in neurocutaneous syndromes, however, there are few data or lack of information on ongoing therapeutic solutions in the majority of the disorder.
Major comments
1. It is not clear for the reviewer why these neurocutaneous syndromes were selected for the manuscript. Please indicate it in the manuscript.
2. Majority of the syndromes are inherited, therefore genetic counseling is of importance, however, it was mentioned only in case of Von-Hippel Lindau syndrome (line 299-300). It is also very important to clearly state the mode of inheritance, since it has an effect on the recurrent risk of the disorder within the family.
3. Where appropriate it should be mentioned that the appearance of certain symptoms may be age related which makes a challenge in the establishment of the diagnosis.
4. It would be easier to follow the manuscript if diagnostic criteria for each disease were displayed in tables. I recommend that these be drawn up.
5. Gene names are displayed incorrect form throughout the manuscript. It should be display in italics.
6. In type 1 Neurofibromatosis patients with microdeletion poses a large group and they have increased risk for malignancies. It should be discuss in the manuscript.
7. In line 54 the authors state: „cutaneous manifestations may present only in affected body regions”. This make no sense. The clinical manifestations always present in the affected body region. Please remove this.
8. In line 60 the displayed reference [11-13] are not appropriate. There is a dedicated publication on diagnostic criteria for NF1 and Legius syndrome. Please change them to “Legius E, Messiaen L, Wolkenstein P, et al. Revised diagnostic criteria for neurofibromatosis type 1 and Legius syndrome: an international consensus recommendation. Genet Med. 2021; 23: 1506–1513. “
9. In line 65 the authors state: “Regular monitoring is recommended for tracking the progression of the patient’s condition and to detect any complications early.” This is a very general statement. Please specify it. e.g. annual whole body MRI is recommended to detect not visible cancerous malformations.
10. The authors should be prepare a table with major and minor manifestations of TSC. The authors discuss it in line 139-145, but in a really general level. Please specify them.
11. Sentences in line 276-281 are a bit confusing. In the case of Von-Hippel-Lindau mutations in VHL gene are exclusively responsible for the disorder. The applied technology for testing this gene is the privilege of the laboratory. If we have a patients resembling for neurofibromatosis, gene panel analysis or comprehensive test is more advisable for the differential diagnostic purposes. It is advisable for the authors to discuss it there.
Minor comments
1. Regarding TSC1 and TSC2 it would be advisable to mention the occurance rate of de novo and inherited mutations.
2. In line 274 “molecular diagnosis” or “genetic testing” mean almost the same. One of is redundant, please remove it.
3. In line 275 the authors state: „heterozygous pathogenic variant of pVHL”. This is incorrect. „pVHL” is for protein. Heterozygous pathogenic variant can be found in genes. Here the „VHL” is the gene symbol.
4. In Figure 5 representing a chromosome picture has no additional meaning. Please redraw this figure
Author Response
Thank you for your comments. The following changes have been made.
Why these diseases were chosen is now included in the introduction. The role of genetic counseling was added to the conclusion. Discussion of age related symptoms was made. Diagnostic criteria of diseases were added. Gene names were fixed and included in italics. Microdeletion of NF1 was added. Removed confusing statements. Criteria of NF1 were included in the newly added table based off the recommended article. Statement about regular monitoring was specified. Table was added for TSC. De novo mutations were included along with removal of genetic testing when molecular diagnosis was mentioned. Corrected statement about pVHL.
Reviewer 2 Report
Comments and Suggestions for Authors
Comments on the Quality of English LanguageSome spelling errors
Author Response
Thank you for all of your feedback. Revisions were made by adding new diagnostic criteria and clarifying confusing language.
Reviewer 3 Report
Comments and Suggestions for Authors
The manuscript entitled "Neurocutaneous Diseases: Diagnosis, Management, and Treatment" is a literature review that attempts to update information about the diagnosis, management, and treatment of the heterogeneous group of disorders characterised by the coexistence of cutaneous and neurological signs and symptoms. The review covers multiple aspects, including the genetic basis, clinical manifestations, diagnostic criteria, and available treatments for disorders such as Neurofibromatosis type 1 and 2, Tuberous Sclerosis Complex, Sturge-Weber Syndrome, Von Hippel-Lindau Disease, Ataxia-Telangiectasia, and Osler-Weber-Rendu. Overall, the manuscript is well organised, but as a reader, I would have preferred the text to follow a similar pattern for each described condition.
Here are some suggestions and feedback:
- Providing epidemiological characterisation for each neurocutaneous disorder would enhance the reader's understanding of whether these conditions are common or rare.
- It's important to distinguish between management and treatment as separate concepts.
- Exploring ongoing clinical trials or promising research areas that could influence the treatment landscape in the future would be valuable.
- Discussing the proactive approach in the field of neurocutaneous disorders could offer insights into preventive strategies.
Additionally, the details in the figures were not clearly readable, and the authors are encouraged to provide higher resolution images.
For specific disorders:
Neurofibromatosis 1 & 2
- line 40: chromosome 17q11.2 instead of chromosome 17 (specify that 17q11.2 is the location of interest instead of the entire chromosome 17).
- line 42: specify the pathogenic variant of the NF1 gene instead of the mutation of the NF1 gene, at least the first time you use the term mutation.
Tuberous Sclerosis
- line 113: what do you mean by ”Genomically”?
- line 120: (figure 2) instead of (figure#)
Sturge-Weber-Syndrome
- lines 169-170: the GNAQ (guanine nucleotide-binding protein, Q polypeptide) and GNA11(guanine nucleotide-binding protein, alpha-1) genes are located on chromosomes 9q21.2 and 19p13.3, respectively. (OMIM)
- line 174: Figure 2 instead of Figure 1.
- line 237: In the sentence: "This mutation causes the degradation of the of the hypoxia-inducible factor (HIF)," there is a repeated "of the."
The suggestions aim to enhance clarity and precision in specific areas.
Author Response
Thank you for all of your feedback. The following revisions have been made.
Chromosomes were specified. Corrected to pathogenic variant. Deleted genomically. Corrected to figure 2. Removed repeats of "of the".
Reviewer 4 Report
Comments and Suggestions for Authors
The authors present a very interesting and transversal topic, carrying neurocutaneous syndromes a wide spectrum of clinical manifestations (thus involving different specialists) and genetic conundrums.
1) Because of their complexity and considering the several articles recently published on these diseases, in my opinion, more detailed descriptions of both clinical and genetic features could be provided.
2) If the title cites "diagnosis, management and treatments", these are the three main areas on which each disease section should focus; however, many paragraphs deeply explain pathogenesis, while diagnostic criteria are only quickly and partially exposed (even in those few diseases in which they are well defined); I advise the Authors to add tables summarizing diagnostic criteria. Finally, treatments are only listed superficially (e.g. Lines 107-108) and would certainly deserve major relevance especially when submitted to a clinical Journal.
3) Being intended as a Review, a “Methods” paragraph is lacking.
4) Please, always specify abbreviations (e.g. AVM, etc).
5) Maybe in line 32 "focused on palliative efforts" sounds too neat, considering that in some of the described syndromes a certain percentage of patients will never develop tumors or life-risking conditions.
6) When listing in NF1 criteria, cutaneous neurofibromas are lacking (L 59) and the article reports the old criteria. Ref 6 should be updated with the newest diagnostic criteria edited in 2021 by Legius et al.
7) As regards treatment for NF1, in my opinion, more emphasis could be reserved for selumetinib, being currently the first and only drug specifically approved for plexiform neurofibromas in children
8) Recently changes have occurred in the classification of Neurofibromatosis type 2; I suggest you call it “NF2-related schwannomatosis”, in order to avoid confusion with NF1 (e.g Lines 93-95 are at risk of causing some misunderstanding) and describe it in a paragraph different from that dedicated to NF1.
9) the genetic background is assuming a more and more predominant role in the management of some phakomatosis, with specific variants associated with a more severe phenotype thus requiring stricter surveillance; it could be of help to cite studies on genotype-phenotype correlation
10) As regards TSC paragraph, I would write on everolimus among the possible treatment, explaining the indications for its use in SEGAs and epilepsy (consider the trials EXIST 1-3)
11) L 236-237: there is a mistake in the explanation, the word "protein" instead of "mutation" should be used
12) L 246-249: the sentences sound contradictory: are cutaneous manifestations in VHL important or just a minor feature?
13) L 250-259 (where you explain what melanocytic naevi and café-au-lait macules are): I suggest writing off this paragraph, considering the scarce value in the discussion; if the authors want to explain CALMs, in my opinion, it could be rather useful in the NF1 chapter.
14) Active components of drugs should be written without a capital letter (e.g. belzutifan instead of Belzutifan, L 287)
15) please, modify Fig 6 reducing the importance given to ranking and adding columns with the main features of the three forms, thus helping readers in differential diagnosis.
Comments on the Quality of English Language- more attention to proper prepositions and punctuation should be paid
- from the 5th chapter on (VHL), English becomes really poor, some sentences are difficult to interpret or confusing and unprecise (e.g. L 229-231, 235, and 283-302)
Author Response
Thank you for all of your feedback. The following changes were made.
Diseases were specified and new tables added talking about diagnostic criteria. Abbreviations were specified. Palliative efforts was resolved. New criteria for NF1 is included in newly added table. NF2 schwannomatosis was changed.
Round 2
Reviewer 1 Report
Comments and Suggestions for Authors
The authors made several changes according to the reviewer’s comments, however, a number of issues remained without correction.
The reference for Table 1 is missing from the text.
Unfortunately the authors did not remove irrelevant references 12-14 from line 75 and did not insert the correct reference here (Legius 2021).
There are several clinical studies on NF1. I miss mentioning these in the manuscript.
Gene names are still not displayed in italics in all cases.
The section between lines 301-306 seems to be a condensed summary of the molecular testing part of https://www.ncbi.nlm.nih.gov/books/NBK1463/ Unfortunately, the information content became distorted and misleading in the manuscript. Please rewrite this section.
In Figure 5 representing a chromosome picture has no additional meaning. Please redraw this figure
Author Response
- Table 1 reference included.
- References removed.
- Authors unsure what clinical studies reviewer wanted included.
- Gene names should be corrected.
- Molecular testing section revised.
- Chromosome removed.
Reviewer 2 Report
Comments and Suggestions for Authors
I would like Authors to address my comments and concerns provided in the first manuscript review with a point-by-point response incorporating suggested literature reviews or necessary discussion.
Comments on the Quality of English LanguageMinor editing of English language required
Author Response
- De novo mutations and 2 hit pattern included.
- Mentioned other domains of neurofibromin.
- A timing course could not be found from the authors are individuals vary in their presentation and we didn't deem this necessary to standardize as it can vary among individuals. However, peripheral and spinal NF1 were included.
- "No cure" was removed.
- NF2 2 hit pattern included.
- Vectors for AT therapy included.
- Prevalence rates have been included.
- Ataxia definition resolved.
- Paper ran through grammar service to eliminate spelling errors.
Reviewer 4 Report
Comments and Suggestions for Authors
Dear Authors,
I appreciate the few new tables summarizing diagnostic criteria for the first 3 diseases and the correction of some typos.
Unfortunately, despite what you wrote in the "author response" section, there is no trace of the word "schwannomatosis in the text and NF1 and NF2 are still associated; no explanation on criteria used for this review are assessed and the paragraphs on possible therapies are still mainly incomplete.
Please, follow my previous recommendations to properly revise the manuscript.
Comments on the Quality of English LanguageNo major changes in English quality have been made, please revise the language mainly from the VHL section on.
Author Response
- NF2 changed to NF2 related schwannomatosis
- This paper was a review and followed the similar format of others paper submitted to this journal that were published without a methods section.
- Therapies were further expanded upon.